# Left Main Coronary Artery Disease—Current Management and Future Perspectives

**DOI:** 10.3390/jcm11195745

**Published:** 2022-09-28

**Authors:** Emil Julian Dąbrowski, Marcin Kożuch, Sławomir Dobrzycki

**Affiliations:** Department of Invasive Cardiology, Medical University of Bialystok, 24A Sklodowskiej-Curie St., 15-276 Bialystok, Poland

**Keywords:** left main coronary artery, percutaneous coronary intervention, coronary artery bypass grafting, coronary artery disease, coronary revascularization

## Abstract

Due to its anatomical features, patients with an obstruction of the left main coronary artery (LMCA) have an increased risk of death. For years, coronary artery bypass grafting (CABG) has been considered as a gold standard for revascularization. However, notable advancements in the field of percutaneous coronary intervention (PCI) led to its acknowledgement as an important treatment alternative, especially in patients with low and intermediate anatomical complexity. Although recent years brought several random clinical trials that investigated the safety and efficacy of the percutaneous approach in LMCA, there are still uncertainties regarding optimal revascularization strategies. In this paper, we provide a comprehensive review of state-of-the-art diagnostic and treatment methods of LMCA disease, focusing on percutaneous methods.

## 1. Background

According to the latest WHO reports, in 2019 ischaemic heart disease (IHD) has strengthened its position as a leading cause of deaths since 2000, accounting for 16% of the world’s total deaths. The rise was especially marked in low-, lower-middle, and upper-middle-income countries. Interestingly, although in high-income countries the number of deaths due to IHD declined, it still remained the main cause of death [1].

Since the early development of coronary artery angiography, it became evident that not all atherosclerotic lesion localizations are equally dangerous. Due to its anatomical features, patients with an obstruction of the left main coronary artery (LMCA) may be at exceptionally high risk. Depending on coronary artery dominance, LMCA supplies blood to 75–100% of the myocardium [2]. Knowing that, there is no wonder that LMCA in the past was known as ‘the artery of sudden death’ [3]. During the early coronarography era, clinicians reported even a 10% risk of death due to LMCA catheterization, and suggested special caution when performing angiography in patients with suspected left main coronary artery disease (LMCAD) [4]. Research available at the time reported over 50% five-year mortality among the patients who received only pharmacological treatment [5]. In the meta-analysis performed by Yusuf et al. 10-year mortality in the group of patients with LMCAD exceeded even the mortality rate of patients with the involvement of three vessels [5].

The poor prognosis of patients with LMCAD gradually improved with the development of revascularization techniques. In the 1970s, coronary artery bypass grafting (CABG) was implemented in the treatment of coronary artery disease (CAD) [6]. Surgical efficacy was proven in observational studies and early randomized clinical trials (RCT), which resulted in wide acknowledgement of this treatment as a method of choice in LMCAD [7]. The following years brought another breakthrough in the treatment of CAD. In 1978, Andreas Gruntzig published a description of five patients not suitable for CABG, including two with LMCAD, successfully treated with a novel method—percutaneous transluminal coronary angioplasty (PTCA) [8]. As he reported few severe acute complications, the initial results were promising. Yet, there was still no data on its long-term complications and safety. Further research revealed that application of PTCA in LMCA was highly unfavorable, and bore a high risk of death and restenosis [9]. It led to the grounding of the CABG position as the first choice for LMCAD treatment for almost twenty years [10].

However, the development of bare-metal stents (BMS) and, finally, drug-eluting stents (DES) led to the necessity of reconsideration percutaneous coronary interventions (PCI) as the method of LMCAD treatment, at least in some subgroups [11]. In the early 2000s, multiple studies provided evidence on the effectiveness and safety of PCI in LMCA, which was eventually reflected in 2009 as the new class IIb recommendation in ACC/AHA guidelines, starting a new chapter in coronary artery revascularization [12,13,14]. Recent years brought several highly acclaimed multicentre RCTs and large-register analyses comparing the use of CABG and PCI in LMCAD [15,16,17,18,19]. Nevertheless, despite the fine quality of the aforementioned research, the long-term outcomes and prognoses of percutaneous treatment of this special disease are inconsistent.

In this article, we aim to provide a comprehensive review of state-of-the-art diagnostic and treatment techniques of left main coronary artery disease, focusing on percutaneous methods.

## 2. Evidence Supporting LMCA Revascularization

Decision on therapeutic strategy in ischaemic heart disease is an important issue. Current guidelines supporting significant LMCA revascularization are based on the early studies documenting the survival advantage of CABG compared to medical therapy (MT). A recent landmark ischemia trial that proved similar effects of an invasive approach compared to MT excluded individuals with significant unprotected left main coronary artery (ULMCA) stenosis [20]. Due to safety concerns, most recent studies addressing invasive and conservative strategies did not include patients with ULMCA. In fact, no RCT directly compared DES with MT in LMCAD. The meta-analysis by Shah et al. that focused on the comparison of CABG to DES to MT, revealed that an invasive approach was associated with better survival over short, intermediate, and long term [7]. To sum up, current evidence supports ULMCA revascularization over MT, however, future ground-breaking RCTs may influence physicians’ approach.

The beginning of the 21st century carried rapid and notable advancements in the field of percutaneous device technology, pharmacological treatment (e.g., dual antiplatelet therapy (DAPT)), procedural techniques, and imaging. The higher risk of adverse cardiovascular events led to the abandonment of BMS in favor of more promising DES [21,22]. All the aforementioned factors have contributed to the renewed enthusiasm for the percutaneous approach in the treatment of LMCAD and resulted in registry studies and, eventually, multicentre RCTs that focused on its comparison with conventional surgical treatment.

### 2.1. Randomized Clinical Trials

Although early registry and nonrandomized studies suggested promising data on outcomes of percutaneous treatment, they were prone to selection bias and confounding. When the results of RCTs targeted on LMCAD treatment were finally published, they mostly suggested the comparable efficacy and safety of percutaneous and surgical treatment in terms of various endpoints [16,17,23,24,25,26,27,28,29,30]. The summary of the major studies and their findings are presented in Table 1. Recent years brought the awaited long-term follow-ups that provided new insights into differences between revascularization strategies [15,16,18,19]. 

Available presently, 10-year follow-up data of LE MANS (Left Main Coronary Artery Stenting), PRECOMBAT (Premier of Randomized Comparison of Bypass Surgery versus Angioplasty Using Sirolimus-Eluting Stent in Patients with Left Main Coronary Artery Disease) and SYNTAXES (SYNTAX Extended Survival) maintained the previously reported trends of comparable outcomes provided by both strategies. It is, however, noteworthy that all the above RCTs were underpowered due to either too small population or unexpectedly low event rates, thus their findings should be considered hypotheses-generating.

Only two of the RCTs focusing solely on LMCAD treatment were sufficiently powered for non-inferiority testing of prespecified major adverse cardiac and cerebrovascular events (MACCE)–EXCEL (Evaluation of Xience Everolimus Eluting Stent vs. Coronary Artery Bypass Surgery for Effectiveness of Left Main Revascularization) and NOBLE (Nordic-Baltic-British Left Main Revascularization) trials. While the highly anticipated results were expected to shed more light on the uncertainties, not only did they not break the deadlock, but also provided conflicting outcomes. Although the added value of studies is indisputable, they both have been criticized for their shortcomings and aroused various controversies. To discuss and understand the dissimilarities in outcomes, it is important to know the analogies and disparities between their design.

They have been both conducted as non-inferiority randomized trials comparing PCI with CABG in LMCAD. The EXCEL trial recruited 1,905 patients with angiographical LMCA stenosis of 70% or 50–70% stenosis with additional non-invasive or invasive testing proving hemodynamically significant lesion. Additional inclusion criterium was low or intermediate anatomical complexity (expressed as SYNTAX (Synergy between Percutaneous Coronary Intervention with TAXUS and Cardiac Surgery) score ≤ 32). Patients were randomly allocated to either PCI (n = 948) or CABG (n = 957) group. All patients received second-generation fluoropolymer-based cobalt–chromium everolimus-eluting stents (EES). The NOBLE trial consisted of 1,201 patients, 598 randomized to PCI and 603 to CABG. Inclusion criteria involved visually assessed stenosis of LMCA ≥ 50% or fractional flow reserve (FFR) ≤ 0.80. SYNTAX score was not utilized as the inclusion or exclusion criterium. Instead, patients with a complex lesion or more than three additional non-complex lesions were excluded (complex lesions were defined as chronic total occlusions, bifurcation lesions requiring two stent techniques, or lesions with calcified or tortuous vessel morphology). In the beginning, around 10% of patients were treated with first-generation sirolimus-eluting stents (SESs), and the rest received newer-generation biolimus-eluting stents (BESs). Primary composite endpoints differed between the two studies: EXCEL included all-cause death, myocardial infarction (MI), and stroke, while NOBLE predefined MACCE as all-cause death, non-procedural MI, stroke, or any repeat revascularization.

Both three-year and five-year follow-up of EXCEL presented that PCI is non-inferior to CABG in terms of primary composite endpoints. However, when analyzing individual endpoints in the five-year follow-up, it turned out that death from any cause occurred more frequently in the PCI group. On the other hand, short- and mid-term follow-up of NOBLE revealed that although PCI was inferior to CABG, all-cause mortality rates were not affected.

At a glance, contrary results of the landmark RCTs require cautious analysis. There are at least a few sources of discrepancies that could be located on the studies’ timelines. At the very beginning, the assessment strategies for eligibility of patients for both revascularization techniques were different. In EXCEL there was a clear heart team (i.e., an interventional cardiologist and a cardiac surgeon) involvement in decision-making, while the multidisciplinary qualification was more ambiguous in NOBLE. It might have led to the heterogeneity of patients enrolled in the study, affecting later outcomes. Secondly, there were previously discussed differences in used device technology. According to the NOBLE authors, the average LMCA diameter is above 4 mm (average 5.7 mm), while the maximum size of used BES was 4.0 mm. Only half of the patients underwent post-dilatation with balloons larger than 4 mm, suggesting that stent underexpansion and malapposition might have contributed to the high rate of revascularizations. If we compare rates of therapy failure defined as definite stent thrombosis or symptomatic graft occlusion between two trials, there are major discrepancies. In NOBLE there were no significant differences in failure ratios between the two strategies (2% vs. 4% for PCI and CABG, respectively), while the superiority of the percutaneous method was apparent in the EXCEL trial (1.1% vs. 6.5% for PCI and CABG, respectively). Additionally, in NOBLE percutaneous treatment in patients with low SYNTAX score (<23) was unexpectedly found to be significantly inferior to surgery. Interestingly, in the PCI arm in NOBLE, there was an unexplainable high prevalence of stroke occurrence at one year, which coincides with DAPT cessation. Arguably the most important disparities concerned composite primary endpoints. EXCEL focused on the previously discussed hard endpoints, while NOBLE adopted any revascularization as well. On the one hand, the NOBLE investigators excluded periprocedural MI from MACCE, but on the other, the EXCEL researchers used the SCAI definition of MI which favored PCI [32]. As a result, NOBLE outcomes were largely driven by the inclusion of revascularization into composite endpoints, and on the other side EXCEL non-inferiority of PCI was driven by a lower incidence of periprocedural MI. Knowing the late cross-over of event curves, the longer follow-up of the studies will deliver further valuable information on both procedures’ effectiveness and safety.

### 2.2. Meta-Analyses

The publication of numerous long-term follow-up outcomes of RCTs prompted the patient- and study-level meta-analysis. Most of them are consistent with each other, stating that PCI and CABG are similarly safe in terms of all-cause mortality, MI, and stroke, however percutaneously treated patients more frequently required repeat revascularization. The summary of selected meta-analyses is presented in Table 2.

### 2.3. Special Groups

When considering the optimal strategy for LMCA revascularization, not only the severity of CAD and the possibility of achieving complete revascularization is important, as comorbidities, age, and past medical history also influence the treatment. As RCTs are prone to strict enrolment criteria, they might not appropriately reflect patients that are met in everyday practice. According to multiple reports, patients over the last years tend to be older and sicker. As a consequence, due to high surgical risk, PCI is more often a method of choice. It was well expressed by Kataruka et al. in their analysis, reporting over a two-fold increase in LMCA PCI between the years 2005 and 2017 [39]. The apparent diversity of patients and little evidence supporting management strategies in special groups may raise clinical uncertainties.

#### 2.3.1. Diabetes Mellitus

Diabetes mellitus (DM) is one of the most challenging comorbidities associated with CAD. Patients with DM are at a higher risk of developing severe CAD, complications after revascularization, and risk of restenosis [40]. After the initial decrease of DM-related complications, such as MI, recent years have brought an alarming trend of their resurgence, mainly among younger patients [41]. Traditionally, diabetes has been a strong indication for CABG treatment, especially in patients suffering from a multivessel disease (MVD). Although there is evidence supporting CABG in such cases, the choice of optimal treatment in LMCAD is blurrier [34]. In the BMS era, there was a noticeable benefit of CABG over PCI in LMCAD with concomitant DM, but the development of DES has once again led to the need for reconsideration of optimal revascularization strategy [42]. Although no trial has solely focused on diabetic patients, there is recent evidence derived from subanalyses of the aforementioned modern RCTs and large registry studies that suggest similar outcomes of PCI compared with CABG. Head et al. in their pooled analysis of individual patient data from 11 trials, reported that five-year all-cause mortality was similar in patients treated with either method, and diabetes status did not interact with the treatment effect (*p* for interaction = 0.13) [34]. A more recent meta-analysis performed by Sabatine et al. supported these findings [38].

In conclusion, PCI with modern-era DES became a valuable option for diabetic patients with LMCAD. Nevertheless, CABG remains the treatment of choice for MVD involving LMCA with concomitant DM.

#### 2.3.2. Chronic Kidney Disease

Chronic kidney disease (CKD) is a well-known condition associated not only with more diffuse CAD, but also with a poorer prognosis [43]. A recently published RCT suggested that there is no benefit of the early revascularization approach in such patients [44]. Data on LMCAD treatment with concomitant CKD is limited, but lately presented evidence mostly supports the equivalence of CABG and PCI, especially in terms of all-cause mortality rates [45,46,47,48]. Patients who obtained percutaneous treatment more often required repeat revascularization, while surgery was linked with a higher risk of stroke. However, the benefits of CABG were significant in severe CKD, which is consistent with Lee et al. findings of a patient-pooled analysis of PCI outcomes [49].

#### 2.3.3. Left Ventricular Dysfunction

Heart failure (HF) and CAD often accompany each other, as the latter is the most common cause of left ventricular (LV) dysfunction [50]. The improved survival of patients with MI, among others, has resulted in the increasing prevalence of HF over the last years [50,51]. According to Bollano et al. the utilization of coronary angiography in patients with HF between 2000 and 2018 has increased by 5.5% per year, resulting in an increased number of revascularizations and a better long-term prognosis. Interestingly, no such increase was seen for angina pectoris and ST-elevation myocardial infarction (STEMI) [52]. The choice of revascularization method in patients with reduced left ventricular ejection fraction (LVEF) may determine their long-term survival. The STICH (Surgical Treatment for Ischemic Heart Failure) trial proved that CABG is superior to medical therapy in patients with ischemic cardiomyopathy at a ten-year follow-up [53]. On the other hand, the most recent REVIVED trial has questioned the reasonableness of PCI in patients with severe ischaemic LV dysfunction [54]. Importantly, the study included 95 patients with LMCAD. In the overall cohort and in the LMCAD subgroup, the percutaneous approach did not result in a lower incidence of death from any cause or hospitalization for heart failure when compared to medical therapy. However, there are a few concerns regarding the study design. First of all, it was an open-label trial. Secondly, lesion-significance assessment did not include intravascular imaging or physiological assessment, which may have especially influenced patients with LMCAD. Thirdly, as much as 66% of individuals in the PCI arm were asymptomatic, therefore results cannot be easily extrapolated to patients with angina. Several observational studies and meta-analyses compare invasive methods of treatment revealed better outcomes of surgery in patients with reduced LVEF and CAD [34,55,56,57,58]. When it comes to the management of patients with LMCAD, Wolff et al. reported that CABG was associated with significantly improved survival compared with PCI [57]. A recent analysis of the IRIS-MAIN (Interventional Research Incorporation Society-Left MAIN Revascularization) registry proved that PCI was inferior to CABG in terms of the primary composite outcome of death, MI, or stroke in patients with LVEF < 45% [59]. On the other hand, the results of the EXCEL trial and Bangalore et al. study showed similar results regarding primary composite endpoint and long-term survival, respectively [24,60]. Moreover, the superiority of the surgical approach was not significant in the IRIS-MAIN registry when complete revascularization was achieved. It is consistent with other studies, indicating that completeness of revascularization should be a priority in patients with reduced LVEF [57,60]. All things considered, contemporary evidence suggests that patients suffering from LMCAD and reduced LVEF may benefit best from CABG. For those ineligible for surgery, complete percutaneous revascularization may be a valuable alternative.

#### 2.3.4. Age

With improving life expectancy, it is projected that in the United States by 2050 will be home to 18 million people aged 85 or above [61]. Age is a powerful risk factor for CAD, adverse outcomes after cardiovascular events, and complications related to invasive treatment [62]. Elderly patients undergoing cardiac surgeries may be at an especially high risk of negative outcomes. Tran et al. in their analysis revealed that frailty syndrome was remarkably more prevalent in the group of patients undergoing CABG compared with patients undergoing major noncardiac surgery (22% vs. 3%). Recent studies comparing the percutaneous approach with surgical treatment of CAD brought mixed results. The superiority of CABG was especially marked in patients with MVD, but not in LMCAD [63,64,65]. A substudy from the DELTA registry (Drug-Eluting stent for LefT main Artery) found no difference in the occurrence of the primary endpoint in octogenarians after CABG and PCI [66]. Recently published results of a subanalysis of the ten-year follow-up SYNTAX Extended Study which focused on elderly individuals (>70 years old) with three-vessel disease and/or LMCAD reported comparable ten-year all-cause death, life expectancy, five-year MACCE, and five-year quality of life (QOL) status irrespective of revascularization mode [67]. Moreover, there was no significant difference between the relative risks of the treatment effects in the EXCEL trial, and no interaction between age and revascularization methods for the primary outcome was found in the NOBLE trial [15,16]. Sabatine et al. in their meta-analysis found no statistically significant heterogeneity for five-year all-cause deaths in a group of patients suffering from LMCAD aged ≥65 compared with <65 years old [38]. Based on present-day evidence, providing similar effects concerning mortality and QOL, PCI is an important alternative to CABG in LMCAD treatment in the elderly. Results suggest that concomitant comorbidities, frailty syndrome, and expected QOL, rather than chronological age, might be more relevant when considering optimal revascularization strategy in this group.

#### 2.3.5. Lesion Anatomy

When planning the optimal revascularization technique for LMCA, lesion localization and vessel anatomy must be taken into consideration. LMCA is usually divided into three segments—ostium, shaft, and distal segment (Figure 1). As atherosclerotic plaques can localize in every part, treatment strategies are different. The early RCT conducted by Boudriot et al. reported that the incidence of MACCE in the PCI arm differed dramatically regarding lesion location (1.0% in ostium/shaft and 18% in distal segment) [29]. Later, MAIN-COMPARE (Revascularization for Unprotected Left Main Coronary Artery Stenosis: Comparison of Percutaneous Coronary Angioplasty Versus Surgical Revascularization) and DELTA registry analyses brought evidence that lesions located in the ostium and shaft treated by either revascularization method provided comparable outcomes [68,69]. Earlier, the analysis of the latter registry revealed that bifurcation compared with ostial/shaft angioplasty was associated with a higher incidence of MACCE [70]. Long-term follow-up of MAIN-COMPARE demonstrated unfavorable outcomes of PCI compared with CABG for distal LMCAD [71]. Percutaneous treatment was associated with a significantly higher risk for death and composite outcome (hazard ratio (HR): 1.78, 95% confidence interval (95% CI): 1.22–2.59; HR: 1.94, 95% CI: 1.35–2.79 for death and composite outcome, respectively). In contrast, this effect was not observed for ostial or shaft lesions. Interestingly, analysis of the EXCEL trial proved only greater rates of repeat revascularizations, with no influence on the incidence of primary composite outcome after PCI, compared with CABG in the group of patients with bifurcation disease. In the case of lesions located in the ostium/shaft, both treatment methods provided similar results in terms of primary composite outcomes and repeat revascularization rates [72]. Recent findings of a meta-analysis performed by De Filippo et al. supported the superiority of CABG in distal but not in ostial/shaft LMCAD [73]. In summary, contemporary evidence suggests that heterogeneity related to the location of atherosclerotic plaques in LMCA is an important factor that should influence the decision regarding revascularization method. For ostial/shaft lesions, both techniques provide similar prognosis and durability, whereas surgery gives better outcomes when applied to the distal LMCAD. In patients with bifurcation disease selected for percutaneous treatment, better outcomes may be achieved by preferably using the double kissing crush (DK crush) stenting technique, the appliance of intravascular imaging, and appropriate stent optimization [74,75].

## 3. State-of-the-Art Evaluation of LMCAD

Significant LMCA stenosis is detected in 4–6% of patients referred for coronary angiography, occasionally also in asymptomatic individuals [4]. Knowing the unfavorable prognosis of untreated LMCAD, precise evaluation of atherosclerotic plaque is essential in further management. Due to overlapping of side branches, lesion eccentricity, vessel foreshortening, and angulation, conventional coronary angiography has its limitations, especially in intermediate (40–70%) LMCA narrowing. Moreover, the significance of stenosis assessed angiographically is observer-dependent, and the reproducibility of results is low even between experienced clinicians [76,77]. To avoid misclassification of the disease, recent years brought the development of various adjunctive tools that are helpful in the decision-making process.

### 3.1. Intravascular Imaging

Intravascular ultrasound (IVUS) is the best-established method of intravascular imaging in LMCAD evaluation. It may provide valuable information on the plaque extent, cross-sectional characteristics of the lesion, and minimal lumen area (MLA) in LMCA and its branches (i.e., left anterior descending artery (LAD), left circumflex artery (LCx)). As it became evident that plaque burden at the MLA is an independent predictor of events, researchers strived to set an optimal threshold for determining the significance of LMCA stenosis [78,79]. Firstly, based on the analysis of 55 patients and a fractional flow reserve (FFR) of 0.75, Jasti et al. proposed a cut-off value of 5.9 mm^2^ [80]. Later, the prospective multicentre LITRO study validated an MLA of 6.0 mm^2^ as a safe value for LMCA revascularization deferral [81]. In a two-year follow-up period, between patients with MLA < 6.0 mm^2^ who underwent revascularization and deferred patients with MLA ≥ 6.0 mm^2^, there were no significant differences in survival and MACCE rates. Since then, the MLA of 6.0 mm^2^ became a widely acknowledged cut-off value for deferring revascularization of the LMCA. Nonetheless, both of the aforementioned studies were conducted in Western populations. Park et al. in their analysis of 112 Asian individuals proposed IVUS derived MLA of 4.5 mm^2^ as a cut-off value for an FFR of ≤0.8 [82]. A plausible explanation of these discrepancies may include ethnic differences in coronary artery dimensions. The mean MLA of patients included in the Asian study was 4.8 mm^2^, while Jasti et al. reported a mean MLA of 7.65 mm^2^ in their study group. Ethnic differences in LMCA anatomy were also supported by a comparative study of 99 Asian and 99 United States white patients (MLA 5.2 ± 1.8 vs. 6.2 ± 14 mm^2^, respectively) [83].

Not only is IVUS a useful tool for LMCAD assessment, but also it may provide important information on stent adequate expansion and apposition. Early insights from the MAIN-COMPARE registry provided evidence on a better prognosis of patients with LMCAD who underwent PCI under the guidance of IVUS in comparison to only conventional angiography [84]. The reduction in three-year incidence of mortality was especially marked in the group of patients who received DES (4.7% vs. 16.0%, log-rank *p* = 0.048) and no difference was observed in the group treated with BMS (8.6% vs. 10.8%, log-rank *p* = 0.35). Further registry studies supported these findings [85,86,87]. The meta-analysis of ten studies performed by Ye et al. revealed that IVUS-guided PCI of LMCA impressively reduced the risks of all-cause death by 40% compared with angiography-guided PCI [88]. The benefit of IVUS-guidance may especially include stent optimization. It was proved in an early analysis of RCTs by Doi et al. that post-intervention minimum stent area (MSA) measured by IVUS was an important factor that could predict in-stent restenosis (ISR) after nine-months of follow-up, and the authors suggested an MSA threshold of 5.7 mm^2^ for paclitaxel-eluting stents [89]. In the EXCEL trial IVUS-substudy there was a strong association between the group of patients with small final MSA (4.4–8.7 mm^2^) and the occurrence of adverse events during long-term follow-up, compared with patients with the largest MSA (11.0–17.8 mm^2^) [90]. The currently best-known proposed MSA cut-off values that predicted ISR are 5.0 mm^2^ for LCx, 6.3 mm^2^ for LAD, 7.2 mm^2^ for confluence zone, and 8.2 mm^2^ for LMCA [91]. However, nowadays some clinicians advocate for higher MSA thresholds, as in the DK-CRUSH VIII trial (>10 mm^2^, >7 mm^2^, >6 mm^2^ for LMCA, LAD, and LCx, respectively) [92]. To sum up, IVUS is an important tool that can improve PCI performance, leading to fewer procedural-related complications and a better prognosis in patients with LMCAD.

Optical coherence tomography (OCT) is a newer method that can provide excellent resolution images influencing a better assessment of plaque phenotype and identification of PCI-related complications. However, due to technology that requires proper blood clearance, OCT cannot be applied to coronary artery ostia. Another drawback includes low tissue penetration which limits the utilization of this method in LMCA stenosis assessment [93]. Despite that, recent studies investigated its outcomes in PCI of LMCA in comparison with IVUS and conventional angiography, especially in bifurcation disease. In the retrospective analysis of 730 patients, OCT was found to be superior to angiography in distal LMCA stenting with no difference compared to IVUS-guidance [94]. In the LEMON trial that analyzed the feasibility, safety, and impact of OCT-guided LMCA PCI, the primary endpoint of procedural success was achieved in 86% of subjects, suggesting that OCT may be a suitable tool for PCI guidance in distal LMCA [95]. Although contemporary results are promising, further research that investigates safety, long-term outcomes in big arteries, and OCT correlation with physiological assessment is needed.

### 3.2. Physiological Assessment

Knowing the limited accuracy of conventional coronary angiography in the evaluation of LMCAD significance, a physiological assessment may deliver crucial information on the ischemic potential of vessel narrowing, determining further management strategy. A study conducted by Hamilos et al. proved that the FFR threshold of ≥0.80 for LMCA revascularization deferral is safe and clinical outcomes in such patients were similar to those who obtained surgical treatment based on the FFR values < 0.80 [96]. The data on the safety and feasibility of FFR-based deferral was later supported by various meta-analyses, RCTs, and register studies [97,98,99,100]. Moreover, decisions based on visually assessed 50% diameter stenosis (DS) may not accurately reflect the hemodynamic and functional significance of the vessel narrowing, especially in LMCA. Interestingly, an analysis of 152 patients revealed that the optimal cut-off value of DS for predicting FFR ≤ 0.80 was 43%, and multiple studies supported visual-functional mismatch in patients with LMCA lesions [96,101,102]. However, it is noteworthy that FFR interpretation in patients with bifurcation disease or downstream stenoses requires special caution, as it may cause under- or over-estimation of LMCA narrowing functional significance [103,104,105].

Apart from the pre-PCI assessment of LMCAD, FFR is also a useful tool in post-PCI functional optimization or jailed side branch management. According to previous studies that focused on functional significance of side branches after bifurcation crossover stenting, angiography alone tends to overestimate the functional severity of stenoses [106,107]. When it comes to LMCA, Lee et al. reported that only 16.9% of patients that underwent simple crossover stenting had FFR < 0.80 in jailed LCx, and no correlation between FFR values and angiographic percent DS was found [108]. Moreover, at five years, patients with higher FFR values had lower target lesion failure (TLF) rates, while no difference in such outcomes was found based solely on DS. It suggests insufficient angiographic accuracy in the evaluation of jailed LCx functional significance and, consequently, that in most cases complex procedures can be avoided by postinterventional FFR assessment.

Recently, instantaneous wave-free ratio (iFR) established the position of a valuable tool that provides outcomes non-inferior to FFR in CAD treatment [109,110,111]. However, data on its safety and long-term clinical outcomes in LMCAD assessment is currently limited. A study by Warisawa et al. indicates that iFR cutoff ≤ 0.89 for LMCA revascularization deferral is safe, and at a median follow-up of 30 months, MACCE rates were similar to patients that underwent invasive management [112]. If confirmed in further studies, iFR may become an important adenosine-free alternative to FFR in LMCAD evaluation.

## 4. Percutaneous Management Techniques

Evolution and pursuit of better clinical outcomes in patients with LMCAD also affected percutaneous management techniques. As mentioned before, atherosclerotic plaques localized in the distal segment of LMCA are related to a higher incidence of MACCE compared with ostial/shaft lesions. Therefore, optimal stenting technique for bifurcation disease was a subject of special interest over the last years.

Early RCTs concerning percutaneous treatment provided data on unfavorable outcomes of the two-stent technique in coronary artery bifurcations, and advocated for provisional stenting (PS) in such cases [113,114]. Contrary to them, DKCRUSH-II (Randomized Study on Double Kissing Crush Technique Versus Provisional Stenting Technique for Coronary Artery Bifurcation Lesions) reported that the DK crush technique in selected patients was associated with lower target lesion revascularization (TLR) and target vessel revascularization (TVR) rates compared with PS [115]. The results of this study evoked scientists’ interest in further research of the optimal percutaneous approach in coronary bifurcation disease, including distal LMCA. Besides RCTs that involved all-comer bifurcation lesions, two of them focused exclusively on LMCA. DKCRUSH-V trial investigated the difference in TLF between patients with LMCAD that underwent PS compared with DK crush [74]. At three years, the two-stent technique was associated with significantly better outcomes (TLF occurred in 16.9% and 8.3% of patients in the PS and DK crush groups (*p* = 0.005), respectively), and the advantage was especially marked in complex lesions. On the other hand, in the EBC MAIN (European Bifurcation Club Left Main Study) patients with true LMCA bifurcation lesions were randomly allocated to a stepwise layered provisional strategy group or systemic dual stent approach [116]. Interestingly, although none of the analyzed methods proved to be significantly superior, a single-stent approach provided numerically better outcomes in terms of primary (and most of the secondary) endpoints. As the main findings differ between studies, a closer look into procedural characteristics may clarify the source of the discrepancy. Firstly, it is noteworthy that an earlier DKCRUSH-III study which focused on differences in clinical outcome between DK crush compared with culotte in distal LMCAD proved that at three years culotte stenting was associated with significantly increased rates of MACCE and stent thrombosis (ST) [117]. Moreover, the most recent network meta-analysis comparing bifurcation techniques that included 8318 patients from 29 RCTs reported that DK crush was superior to PS and other two-stent techniques [118]. Yet, in EBC MAIN culotte was the most common, and, on the contrary, DK crush was the least common technique used in a two-stent approach (53% and 5% for culotte and DK crush, respectively). Secondly, the PS protocol differed between the two studies—in EBC MAIN kissing balloon inflation (KBI) of the side vessel after stenting was a part of the procedure, whereas in DKCRUSH-V KBI was permitted only if residual DS of the side branch was >75%, or dissection ≥ type B, or Thrombolysis In Myocardial Infarction (TIMI) flow grade < 3 was present. As the studies comparing KBI with no-KBI in a one-stent approach provide non-consistent results, this difference in protocols presumably influenced the final outcomes of the aforementioned RCTs [119]. Lastly, it is noteworthy that operators included in the DKCRUSH-V study had to be well-experienced, as it was confirmed by sending three to five cases to the trial steering committee, which to some extent might have driven favorable DK crush outcomes. The question of whether if the procedure protocols had been unified would the outcomes of both RCTs be similar is thought-provoking, and suggests that further research with the state-of-the-art approach is needed.

When it comes to the recommendations, a provisional strategy followed by a proximal optimization technique (POT) is preferred for the majority of patients, especially without a true distal LMCA lesion [120]. In case of too distal balloon positioning during POT, carina shift resulting in side branch ostium lumen reduction might occur. In such cases, as described before, FFR assessment of functional significance might be applicable. Importantly, if a suboptimal effect was achieved or complications occur, such an approach allows conversion to a two-stent technique (T-, T and protrusion (TAP) or culotte) for a better final outcome. When deciding between a one-stent and up-front two-stent strategy, the complexity of LMCA lesion should be the key-determinant. Although no universal definition of complexity has been established, developed in the DEFINITION study (Definitions and impact of complEx biFurcation lesIons on clinical outcomes after percutaNeous coronary IntervenTIOn using drug-eluting steNts) criteria are the most acknowledged [121]. Recent results of the DEFINITION II trial, including 28.8% of patients with distal LMCAD, proved that for the pre-specified coronary bifurcation lesions, the complexity criteria two-stent approach was associated with significantly better outcomes compared with PS [122]. Of note, in this study, as much as 77.8% of patients in the two-stent group were treated with the DK crush technique. Current ESC/EACTS guidelines indicate that in true bifurcation lesions of LMCA, DK crush may be preferred over provisional T-stenting (class IIb recommendation, level of evidence B) [123]. Even though presumably superior to other methods, it should be kept in mind that DK crush is technically demanding and optimal effects may be achieved in hands of experienced operators. Selected PCI bifurcation techniques are presented in Figure 2.

## 5. Current Guidelines and Future Directions

Surgical revascularization has established its position as a gold standard for LMCA revascularization, reflected in class I recommendation by European and US guidelines [123,124]. Over recent years, great progress has been made in the field of percutaneous CAD treatment, and its equivalence with CABG in selected patients was supported by gradually accumulated evidence, eventually earning a part in recommendations for LMCAD treatment. In the most recent European guidelines class of recommendation for PCI in LMCA was dependent on SYNTAX score: tertiles–I in the lowest, IIa in intermediate, or III in the highest [123]. On the other hand, last year, updated US clinical practice guidelines gave more unified class IIa recommendation for percutaneous treatment in selected patients for whom PCI can provide equivalent revascularization to that possible with CABG, without anatomical complexity stratification [124]. However, recommendations are consistent with each other when it comes to the multidisciplinary heart team involvement in the decision-making process. Such an approach can improve outcomes and minimize the risk of inappropriate use of revascularization strategies, as a marked variability in PCI-to-CABG ratios between countries was observed [125]. Surgical risk scores, such as the European System for Cardiac Operative Risk Evaluation (EuroSCORE II) and the Society of Thoracic Surgeons (STS) score, and anatomical complexity SYNTAX score may provide useful information that influences heart team discussion toward a more patient-orientated decision. If the outcomes are expected to be comparable, the preferences of the patient should be forefront. The summary of indications for PCI and CABG are presented in Figure 3.

Although great effort has been put to improve outcomes and dispel doubts concerning the optimal approach in LMCAD, not all issues have been resolved. Firstly, knowing the late cross-over of event curves, long- and very long-term follow-up of NOBLE and EXCEL are likely to deliver more information on the actual durability and effectiveness of percutaneous and surgical treatment. Secondly, more subgroup-dedicated studies that investigate optimal treatment options in specific patients are needed. Knowing the unequal clinical outcomes of various stenting techniques and the influence of adjunctive tools on PCI results, contemporary state-of-the-art percutaneous treatment comparison with CABG might start a new chapter in LMCAD revascularization.

## 6. Conclusions

The last decades’ developments and further progress in coronary revascularization methods were arguably one of the greatest steps in cardiology and changed the dramatic course of CAD. Although the declining trend in deaths due to IHD in high-income countries is caused by many factors, technical improvements and widespread access to percutaneous treatment are undoubtedly one of them. As for once-deadly LMCA stenosis, nowadays two effective treatment options are available. There is no unified algorithm for decision-making in LMCAD, but careful selection of patients and a multi-disciplinary heart team approach can provide the best management option at the time. Since the modern coronary revascularization philosophy has become patient-orientated, it is important to emphasize that PCI and CABG are not contradictory to each other but rather complementary in terms of reaching favorable outcomes in various clinical settings. Further years are expected to bring more research on LMCA treatment, but due to constant improvements in both techniques they will likely not break the deadlock and the optimal approach will remain a moving target.

## Figures and Tables

**Figure 1 jcm-11-05745-f001:**
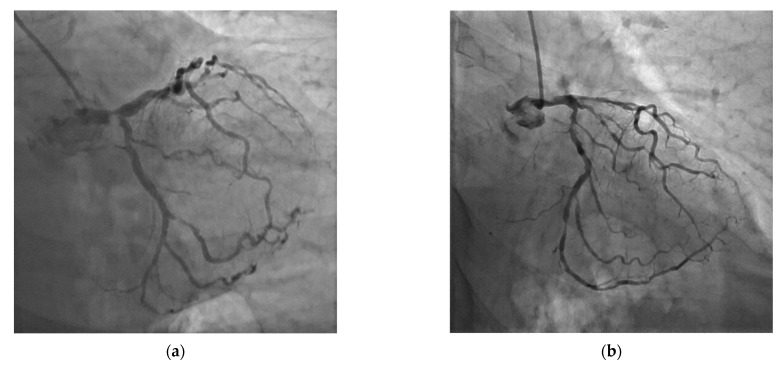
Coronary angiograms. (**a**) Severe distal left main coronary artery (LMCA) lesion. (**b**) Disseminated coronary artery disease with shaft LMCA lesion.

**Figure 2 jcm-11-05745-f002:**
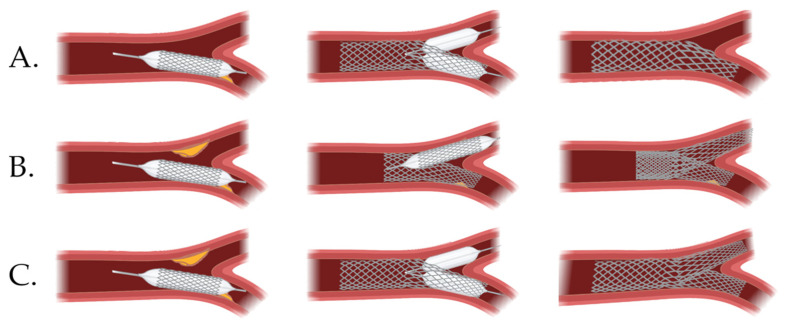
Selected percutaneous coronary intervention bifurcation techniques. (**A**) Provisional stenting, (**B**) culotte, (**C**) double kissing crush.

**Figure 3 jcm-11-05745-f003:**
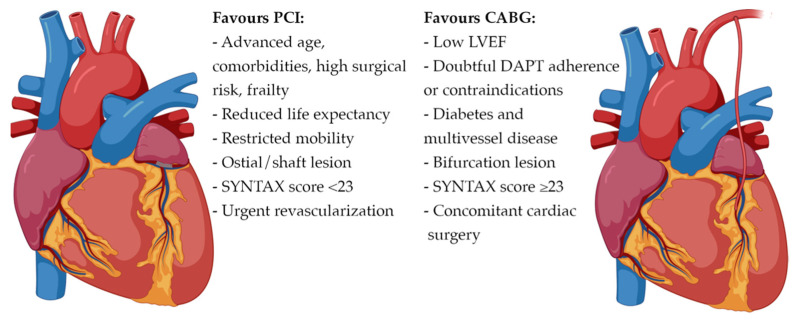
Indications for percutaneous coronary intervention and coronary artery bypass grafting in left main coronary artery disease. CABG–coronary artery bypass grafting, DAPT–dual antiplatelet therapy, LVEF–left ventricular ejection fraction, PCI–percutaneous coronary intervention, SYNTAX - Synergy Between Percutaneous Coronary Intervention With TAXUS and Cardiac Surgery.

**Table 1 jcm-11-05745-t001:** Summary of random clinical trials comparing PCI with CABG in left main coronary artery disease.

	LE MANS [17]	Boudriot et al. [29]	SYNTAX-LM [27,31]	PRECOMBAT [19,25]	EXCEL [15,24]	NOBLE [16,23]
Recruitment period	2001–2004	2003–2009	2005–2007	2004–2009	2010–2014	2008–2015
Follow-up (years)	10	1	5; 10 for mortality only	10	5	5
PCI/CABG (n)	52/53	100/101	357/348	300/300	948/957	592/592
Bifurcation disease (%)	58	72	61	65	81	81
Mean LVEF (%)	54	65	N/D	61	57	60
Age (years)	61	68	65	62	66	66
IVUS (%)	Recommended	No recommendation	N/D	91	77	74
Mean SYNTAX score	N/D	23	30	25	21	22
Stents	BMS and DES (35%)	DP-SES	DP-PES	DP-SES	DP-EES	BP-BES and DP-SES (8%)
OPCAB (%)	1.9	46	N/D	64	29	16
LIMA (%)	72	99	97	94	99	96
Primary endpoint	Change in LVEF	All-cause death, MI, repeat revascularization	All-cause death, stroke, MI, repeat revascularization; 10-years all-cause death	Any-cause death, MI, stroke, TVR	Any-cause death. MI, stroke	Any-cause death, nonprocedural MI, stroke, repeat revascularization
Outcomes	Trend toward higher LVEF in PCI	PCI inferior to CABG	PCI non-inferior to CABG at 5-years; No difference in all-cause death at 10-years	PCI non-inferior to CABG	PCI non-inferior to CABG	PCI inferior to CABG

BMS—bare metal stents, BP-BES—biodegradable polymer biolimus-eluting stent, CABG—coronary artery bypass grafting, DES—drug-eluting stent, DP-EES—durable polymer everolimus-eluting stent, DP-PES—durable polymer paclitaxel-eluting stent, DP-SES—durable-polymer sirolimus-eluting stent, EXCEL—Evaluation of Xience Everolimus Eluting Stent vs. Coronary Artery Bypass Surgery for Effectiveness of Left Main Revascularization, IVUS—intravascular ultrasound, LE MANS—Left Main Stenting, LIMA—left internal mammary artery, LVEF—left ventricular ejection fraction, MI—myocardial infarction, N/D—no data, NOBLE—Nordic-Baltic-British Left Main Revascularization, OPCAB—off-pump coronary artery bypass, PCI—percutaneous coronary intervention, PRECOMBAT—Premier of Randomized Comparison of Bypass Surgery versus Angioplasty Using Sirolimus-Eluting Stent in Patients with Left Main Coronary Artery Disease, SYNTAX—Synergy Between Percutaneous Coronary Intervention With TAXUS and Cardiac Surgery, SYNTAX-LM—left main substudy of the SYNTAX, TVR—target vessel revascularization.

**Table 2 jcm-11-05745-t002:** The summary of selected meta-analyses of RCTs comparing PCI with CABG in LMCAD.

	Palmerini et al. [33]	Head et al. [34]	Ahmad et al. [35]	Bajraktari et al. [36]	D’Ascenzo et al. [37] *	Sabatine et al. [38]
Year of publication	2017	2018	2020	2020	2021	2021
Number of analyzed RCTs	6	11	5	5	4	4
Number of patients (PCI/CABG)	4686 (2347/2339)	4478 (2233/2245)	4612 (2303/2309)	4499 (2249/2250)	4394 (2197/2197)	4394 (2197/2197)
Primary outcome	All-cause mortality	All-cause mortality	All-cause mortality	A composite of all-cause mortality, MI, or stroke	All-cause mortality	All-cause mortality
Results for primary outcome	HR = 0.99, 95% CI 0.76–1.3, *p* = 0.74	RR = 1.07, 95% CI 0.87–1.33, *p* = 0.52	RR = 1.03, 95% CI 0.82–1.30, *p* = 0.78	RR = 1.13, 95% CI 0.94–1.36, *p* = 0.19	OR = 0.93, 95% CI 0.71–1.21, *p* = 0.58	HR = 1.10, 95% CI 0.91–1.32, *p* = 0.33
Other findings	CV mortality: HR = 1.01, 95% CI 0.72–1.42, *p* = 0.83 MI: HR = 1.33, 95% CI 0.84–2.11, *p* = 0.11 Stroke: HR = 0.71, 95% CI 0.34–1.49, *p* = 0.31 UR: HR = 1.74, 95% CI 1.47–2.07, *p* < 0.001 Significant interaction for CV mortality between treatment and the SYNTAX score, *p* for interaction = 0.03	In diabetic patients: RR = 1.34, 95% CI 0.93–1.91, *p* = 0.11; In non–diabetic patients: RR = 0.94, 95% CI 0.72–1.23, *p* = 0.65, *p* for interaction = 0.13 SYNTAX score 0–22: RR = 0.91, 95% CI 0.60–1.36, *p* = 0.64 SYNTAX score 23–32: RR = 0.92, 95% CI 0.65–1.30, *p* = 0.65 SYNTAX score ≥ 33: RR = 1.39, 95% CI 0.94–2.06, *p* = 0.10, *p* for interaction = 0.38	CV mortality: RR = 1.03, 95% CI 0.79–1.34, *p* = 0.82 Stroke: RR = 0.74, 95% CI 0.36–1.50, *p* = 0.40 MI: RR = 1.22, 95% CI 0.96–1.56, *p* = 0.11 UR: RR = 1.73, 95% CI 1.49–2.02, *p* < 0.001	All-cause mortality: RR = 1.07, 95% CI 0.89–1.28, *p* = 0.48 CV mortality: RR 1.13, 95% CI 0.89–1.43, *p* = 0.31 Stroke: RR = 0.87, 95% CI 0.62–1.23, *p* = 0.42 MI: RR = 1.48, 95% CI 0.97–2.25, *p* = 0.07 UR: RR = 1.70, 95% CI 1.34–2.15, *p* < 0.001	MACCE (all–cause mortality, MI, stroke, repeat revascularization): OR = 0.69, 95% CI 0.60–0.79, *p* < 0.001 CV mortality: OR = 0.95, 95% CI 0.68–1.32, *p* = 0.75 Stroke: OR = 1.17, 95% CI 0.59–2.31, *p* = 0.66 MI: OR = 0.48, 95% CI 0.36–0.65, *p* < 0.001 Repeat revascularization: OR = 0.53, 95% CI 0.45–0.64, *p* < 0.001	CV mortality: HR = 1.07, 95% CI 0.83–1.37, *p* = 0.61 Stroke: HR = 0.84, 95% CI 0.59–1.21, *p* = 0.36 Spontaneous MI: HR = 2.35, 95% CI 1.71–3.23, *p* < 0.001 Repeat revascularization: HR = 1.78, 95% CI 1.51–2.10, *p* < 0.001 10–year all–cause death: HR = 1.10, 95% CI 0.93–1.29, *p* = 0.25
Interpretation	PCI and CABG showed similar mortality; interaction effect suggesting relatively lower mortality with PCI in patients with low SYNTAX score and relatively lower mortality with CABG in patients with high SYNTAX score	PCI and CABG showed similar mortality, regardless of diabetic status and SYNTAX score	PCI and CABG showed similar mortality; UR was less common after CABG	PCI and CABG showed similar mortality; UR was less common after CABG	PCI and CABG showed similar mortality; CABG reduced risk of MI, revascularization and MACCE, especially in older patients and with high SYNTAX score	PCI and CABG showed similar mortality; MI and repeat revascularization were less common after CABG

* All ORs are reported for CABG compared with PCI, CABG–coronary artery bypass grafting, CI—confidence interval, CV—cardiovascular, HR—hazard ratio, MACCE—major adverse cardiac or cerebrovascular event, OR—odds ratio, PCI–percutaneous coronary intervention, RR—risk ratio, UR—unplanned revascularization.

## Data Availability

Not applicable.

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
