# Peer review of "Left Main Coronary Artery Disease—Current Management and Future Perspectives"

_jcm, 2022, doi:10.3390/jcm11195745_

Round 1

Reviewer 1 Report

This study sought to provide an overview on the management of left main coronary artery disease (LMCAD). The authors synthesized data on the adequate revascularization strategy in case of LMCAD and propose a state of the art on the specificities of diagnostic approach. A focus on PCI techniques for LMCAD is also proposed.

The study is well written and addresses an important issue given the ongoing debate on therapeutic strategy in ischemic heart disease and the tremendous challenge represented by LM disease. The authors should be commended for this work.

I have the following major concerns and comments:

1. Citations in the Introduction section should be updated, many are refered to papers published since 40 years

2. The authors mainly focused the discussion on the revascularization strategy for LMCAD. Given the ongoing debate on therapeutic strategy in ischemic heart disease in general (revascularization vs medical management), could you more deeply discuss the data supporting the need of revascularization in the specific case of LMCAD?

3. Section “2.3.3 Left ventricular dysfunction”: I would suggest to discuss the results of REVIVED trial just published in NEJM (DOI: 10.1056/NEJMoa2206606) which included patients with LMCAD

4. Since the authors focused on PCI techniques for LMCAD, I would suggest to provide a figure illustrating the main techniques presented (provisional, DK crush, etc.).

I have also some minor concerns:

1. Lines 43 and 76: please correct “random” to randomized”

2. Line 324: “not infrequently in symptomless individuals” : this sentence is difficult to understand. Please reformulate

Author Response

This study sought to provide an overview on the management of left main coronary artery disease (LMCAD). The authors synthesized data on the adequate revascularization strategy in case of LMCAD and propose a state of the art on the specificities of diagnostic approach. A focus on PCI techniques for LMCAD is also proposed.

The study is well written and addresses an important issue given the ongoing debate on therapeutic strategy in ischemic heart disease and the tremendous challenge represented by LM disease. The authors should be commended for this work.

Response: We would like to thank the Reviewer for the time and effort spent on reviewing our manuscript. We have considered these comments very carefully and addressed all of them down below.

I have the following major concerns and comments:

  1. Citations in the Introduction section should be updated, many are refered to papers published since 40 years

Response 1: Thank you for this comment. We have updated citations in the Introduction section. However, we found two of the references unreplaceable as they were used to present a historical outline and present current problems in left main coronary management as a consequence of the past research.

  1. The authors mainly focused the discussion on the revascularization strategy for LMCAD. Given the ongoing debate on therapeutic strategy in ischemic heart disease in general (revascularization vs medical management), could you more deeply discuss the data supporting the need of revascularization in the specific case of LMCAD?

Response 2: Thank you for noticing this important issue. We have discussed this matter in the section ‘2. Evidence supporting LMCA revascularization’ of the updated manuscript.

  1. Section “2.3.3 Left ventricular dysfunction”: I would suggest to discuss the results of REVIVED trial just published in NEJM (DOI: 10.1056/NEJMoa2206606) which included patients with LMCAD

Response 3: Thank you for this suggestion. We have discussed the results of this trial in the updated version of the manuscript.

  1. Since the authors focused on PCI techniques for LMCAD, I would suggest to provide a figure illustrating the main techniques presented (provisional, DK crush, etc.).

Response 4: As suggested, in the section ‘4. Percutaneous management techniques’ we have provided the figure illustrating the main techniques mentioned above.

I have also some minor concerns:

  1. Lines 43 and 76: please correct “random” to randomized”

Response 1: Thank you for this comment. We corrected this mistake.

  1. Line 324: “not infrequently in symptomless individuals” : this sentence is difficult to understand. Please reformulate

Response 2: As suggested, this sentence has been revised.

Reviewer 2 Report

In this review Dabrowski and colleagues present their thoughts about Current Management and Future Perspectives of Left Main Coronary Artery Disease (LMCAD) on the basis of the current literature in a very comprehensive way. 

The paper is well written including all the recent knowledge available in this field. The conclusion that there is no unified algorithm for decision-making in LMCAD, but careful selection of patients in a multi-disciplinary heart-team approach reflects the balanced arguments for the surgical and interventional approach throughout the paper. All in all, this paper sums up all one needs to know to get a good overview into the field of LMCAD and its treatment options. The only point of criticism points to English language and style where moderate changes are required. 

Author Response

In this review Dabrowski and colleagues present their thoughts about Current Management and Future Perspectives of Left Main Coronary Artery Disease (LMCAD) on the basis of the current literature in a very comprehensive way. 

The paper is well written including all the recent knowledge available in this field. The conclusion that there is no unified algorithm for decision-making in LMCAD, but careful selection of patients in a multi-disciplinary heart-team approach reflects the balanced arguments for the surgical and interventional approach throughout the paper. All in all, this paper sums up all one needs to know to get a good overview into the field of LMCAD and its treatment options. The only point of criticism points to English language and style where moderate changes are required. 

Response: We are very grateful for the time and effort invested in reviewing our manuscript as well as the kind words. We have edited the manuscript accordingly to the Reviewer’s suggestions. As suggested, the entire manuscript has been thoroughly revised language-wise.